# Serotonin 5-HT_4_ Receptor Agonists Improve Facilitation of Contextual Fear Extinction in an MPTP-Induced Mouse Model of Parkinson’s Disease

**DOI:** 10.3390/ijms20215340

**Published:** 2019-10-26

**Authors:** Toshiaki Ishii, Ken-ichi Kinoshita, Yoshikage Muroi

**Affiliations:** Department of Basic Veterinary Medicine, Obihiro University of Agriculture and Veterinary Medicine, Obihiro, Hokkaido 080-8555, Japan; k.under-the-tree.6060@ezweb.ne.jp (K.-i.K.); muroi@obihiro.ac.jp (Y.M.)

**Keywords:** Parkinson’s disease, MPTP, 5-HT_4_ receptor agonist, memory extinction, dopamine, hippocampus, median raphe nucleus (MnRN), substantia nigra pars compacta (SNpc), reticular part of the substantia nigra (SNr)

## Abstract

Previously, we found that 1-methyl-4-phenyl-1,2,3,6-tetrahydropyridine (MPTP)-induced Parkinson’s disease (PD) model mice (PD mice) showed facilitation of hippocampal memory extinction via reduced cyclic adenosine monophosphate (cAMP)/cAMP-dependent response element-binding protein (CREB) signaling, which may cause cognitive impairment in PD. Serotonergic neurons in the median raphe nucleus (MnRN) project to the hippocampus, and functional abnormalities have been reported. In the present study, we investigated the effects of the serotonin 5-HT_4_ receptor (5-HT_4_R) agonists prucalopride and velusetrag on the facilitation of memory extinction observed in PD mice. Both 5-HT_4_R agonists restored facilitation of contextual fear extinction in PD mice by stimulating the cAMP/CREB pathway in the dentate gyrus of the hippocampus. A retrograde fluorogold-tracer study showed that γ-aminobutyric acid-ergic (GABAergic) neurons in the reticular part of the substantia nigra (SNr), but not dopaminergic (DAergic) neurons in the substantia nigra pars compacta (SNpc), projected to serotonergic neurons in the MnRN, which are known to project their nerve terminals to the hippocampus. It is possible that the degeneration of the SNpc DAergic neurons in PD mice affects the SNr GABAergic neurons, and thereafter, the serotonergic neurons in the MnRN, resulting in hippocampal dysfunction. These findings suggest that 5HT4R agonists could be potentially useful as therapeutic drugs for treating cognitive deficits in PD.

## 1. Introduction

Parkinson’s disease (PD) is a neurodegenerative disease caused by the progressive loss of dopaminergic (DAergic) neurons in the substantia nigra pars compacta (SNpc). Patients with PD show extrapyramidal motor dysfunctions caused by a reduction in DAergic fibers from the SNpc in the striatum [1]. In addition, non-motor symptoms, including olfactory deficits, anxiety, and cognitive impairment, also occur in patients with PD, sometimes appearing before the onset of motor dysfunction [2,3]. These non-motor symptoms, especially cognitive impairment, appear in about 40% of patients with PD [4]. However, the etiology of cognitive impairment in PD remains unknown.

Recently, we reported that 1-methyl-4-phenyl-1,2,3,6-tetrahydropyridine (MPTP)-induced PD model mice (PD mice) showed facilitation of hippocampal memory extinction via reduced cyclic adenosine monophosphate (cAMP)/cAMP-dependent response element-binding protein (CREB) signaling in the hippocampus, which may cause cognitive impairment in PD [5,6]. Moreover, we also demonstrated that administration of rolipram, a phosphodiesterase IV (PDE IV) inhibitor, restores the facilitation of memory extinction in PD mice by stimulating the cAMP/CREB pathway in the hippocampus [6]. Intracellular cascades involving cAMP lead to activation of protein kinase A (PKA) and phosphorylation of CREB, which regulate both hippocampus-dependent learning and the extinction of conditioned fear [7,8]. Moreover, CREB signaling is also importantly involved in both memory extinction and strengthening upon retrieval [9]. However, the reason why hippocampal cAMP/CREB signaling is reduced in PD mice remains unclear.

The serotonin 5-HT_4_ receptor (5-HT_4_R), a Gs protein-coupled receptor, is implicated in pathophysiological events such as depression, anxiety, and cognitive impairment, as well as memory loss in Alzheimer’s disease and a variety of physiological functions [10,11,12]. The 5-HT_4_R is highly expressed in the hippocampus [13], and 5-HT_4_R stimulation leads to increased hippocampal pyramidal cell activity and serotonin release in hippocampal slices [14]. Indeed, the administration of 5-HT_4_R agonists affects both behavioral and biochemical responses, such as an increase in CREB phosphorylation in the hippocampus, and results in the improvement of several pathophysiological events [15,16,17]. On the other hand, it has been reported that serotonergic neurons in the median raphe nucleus (MnRN), which project to the dorsal hippocampus, are impaired in patients with PD [18,19]. In the present study, to investigate whether hippocampal 5-HT_4_R is involved in the facilitation of memory extinction observed in PD mice, we examined the effects of the 5-HT_4_R agonists prucalopride and velusetrag on contextual fear extinction in PD mice. Moreover, we examined the presence of neuronal projection from the substantia nigra (SN) to the MnRN, the serotonergic neurons of which are known to project to the dorsal hippocampus, in consideration of a candidate neuronal pathway from the SN to the hippocampus [20].

## 2. Results

### 2.1. Administration of the 5-HT_4_ Serotonin Receptor Agonists to PD Mice Improved Facilitation of Contextual Fear Extinction

Previously, we demonstrated that the administration of rolipram, a PDE IV inhibitor, restores the facilitation of memory extinction in PD mice by stimulating the cAMP/CREB pathway in the hippocampus [6]. On the other hand, 5-HT_4_R is highly expressed in the hippocampus, the stimulation of which leads to increased hippocampal pyramidal cell activity and serotonin release via activation of the cAMP/PKA pathway [13,14]. Therefore, we investigated whether prucalopride, a 5-HT_4_ serotonin receptor agonist, could improve the facilitation of fear extinction in PD mice. Systemic administration of prucalopride (1.5 or 3 mg/kg, an intraperitoneal injection (i.p.)) 2 h before extinction training prevented the facilitation of contextual fear extinction in PD mice at days 2 and 3 in a dose-dependent manner, but did not have any significant effect on contextual fear extinction in control mice (Figure 1). We also examined the effect of another 5-HT_4_ serotonin receptor agonist, velusetrag, on the facilitation of fear extinction in PD mice. The administration of velusetrag (3 mg/kg, i.p.) also significantly improved the facilitation of contextual fear extinction in PD mice (Figure 2). On the other hand, the two 5-HT_4_R agonists did not improve the impaired rotarod performance in PD mice (data not shown).

### 2.2. Hippocampal mRNA Expression Levels of 5-HT_4_R in Control and PD Mice

5-HT_4_R is highly expressed in the hippocampus [13]. The expression of 5-HT_4_R has been reported to be altered in a rodent model of PD [21]. Therefore, we examined the hippocampal messenger RNA (mRNA) expression level of 5-HT_4_R in PD mice and whether administration of 5-HT_4_R agonists would alter mRNA expression levels using RT-qPCR. The results showed that hippocampal mRNA expression levels of 5-HT_4_R were not significantly different between control and PD mice or among all groups, even after the administration of the 5-HT_4_R agonists (Figure 3).

### 2.3. Analysis of Neuronal Projections from the SN to the MnRN

Fluorogold (FG) injected into the MnRN (Figure 4A(a,b)) was detected in the glutamic acid decarboxylase 67 (GAD67)-positive neurons in the reticular part of the substantia nigra (SNr) (Figure 4C(a,b)), but was not detected in the tyrosine hydroxylase (TH)-positive neurons in the SNpc (Figure 4B(a,b)). These results suggest that γ-aminobutyric acid-ergic (GABAergic) neurons in the SNr project their neuronal terminals to MnRN serotonergic neurons. These findings are in agreement with those reported by Dorocic et al. [22].

### 2.4. Administration of the 5-HT_4_ Serotonin Receptor Agonists to PD Mice Restored the Decrease in cAMP Levels in the Hippocampus

Previously, we have reported that hippocampal cAMP levels decrease in MPTP-treated compared with control mice [6]. Because the 5-HT_4_R receptor is a Gs protein-coupled receptor, the administration of prucalopride or velusetrag may increase hippocampal cAMP levels in MPTP-treated mice. Therefore, we examined the effects of prucalopride and velusetrag on hippocampal cAMP levels. An enzyme-linked immunosorbent assay showed that both prucalopride and velusetrag significantly restored the reduced cAMP levels in PD mice (Figure 5). The effect of velusetrag was stronger than that of prucalopride at the same dose (3.0 mg/kg). On the other hand, the administration of velusetrag, but not prucalopride, significantly increased hippocampal cAMP levels in control mice (Figure 5).

### 2.5. Administration of the 5-HT_4_ Serotonin Receptor Agonists to PD Mice Restored the Decrease in the Number of Phosphorylated CREB (p-CREB)-Positive Cells in the Hippocampal Dentate Gyrus (DG)

Previously, we have reported that the number of p-CREB-positive cells in the hippocampal DG decrease in PD mice, which probably the result of reduced cAMP levels [6]. We therefore examined the effects of prucalopride and velusetrag on the number of p-CREB-positive cells in the hippocampal DG. The administration of the 5-HT_4_R agonists to PD mice significantly restored the number of p-CREB-positive cells in the hippocampal DG to the level in control mice (Figure 6).

## 3. Discussion

Dysfunction of the serotonin system has been implicated in patients with PD [23]. 5-HT_4_R, a Gs protein-coupled receptor, is highly expressed in the hippocampus [13], and is implicated in pathophysiological events such as depression, anxiety, and cognitive impairment [11,15,17]. In the present study, we demonstrated that the 5-HT_4_R agonists prucalopride and velusetrag improved the facilitation of contextual fear extinction in the PD mice without affecting the mRNA expression levels of hippocampal 5-HT_4_R. These results suggest that 5-HT_4_R agonists could be potentially useful as a therapeutic drug for treating cognitive deficits in PD. 

Hippocampal cAMP/CREB signaling is one of the critical signaling cascades for consolidation, re-consolidation, and fear memory extinction. Indeed, many reports on the role of cAMP/CREB signaling in fear memory extinction have been published, with the results showing that an increased level of cAMP enhances retention and slows down extinction of conditioned fear [24], that transgenic inhibition of PKA facilitates fear extinction [25], and that p-CREB expression in the hippocampus increases in fear memory extinction [26]. Previously, we demonstrated that reduced cAMP/CREB signaling in the DG leads to the facilitation of memory extinction in PD mice, because the administration of rolipram, a PDE IV inhibitor, restored the facilitation of memory extinction in PD mice via stimulating the cAMP/CREB pathway in the hippocampus [6]. Moreover, in the present study, the 5-HT_4_R agonists prucalopride and velusetrag restored the decrease in cAMP levels and the reduced number of p-CREB-positive cells in the hippocampal DG of PD mice (Figure 5 and Figure 6). These findings suggest that reduced cAMP/CREB signaling in the DG might be involved in a cause of cognitive impairment in PD mice. However, the reason why hippocampal cAMP/CREB signaling is reduced in PD mice remains unknown. On the other hand, administration of velusetrag (3.0 mg/kg) to control mice significantly increased hippocampal cAMP levels (Figure 5), but did not affect contextual fear extinction, and maintained it (Figure 2). These results suggest that cAMP contents higher than a certain threshold level in the hippocampal neurons might be necessary for normal extinction. 

Serotonergic neurons in the MnRN, which project to the dorsal hippocampus, are impaired in patients with PD [18,19]. We considered the possibility that the degeneration of SNpc DAergic neurons in PD mice may indirectly lead to dysfunction of serotonergic neurons in the MnRN, resulting in an influence on hippocampal function. Therefore, we examined neuronal inputs to serotonergic neurons in the MnRN from the substantia nigra (SN). Figure 4 shows that GABAergic neurons in the SNr, but not DAergic neurons in the SNpc, project their neuronal terminals towards the MnRN serotonergic neurons. The results obtained from our retrograde tracing experiment were identical to those from a trans-synaptic tracing technique using modified rabies viruses [22]. These findings imply that the degeneration of SNpc DAergic neurons might affect GABAergic neurons in the SNr close to the SNpc, which in turn, may influence serotonergic neurons in the MnRN receiving GABAergic neurons from the SNr, resulting in hippocampal dysfunction (Figure 7). On the other hand, mRNA expression levels of 5-HT_4_R were not significantly different between control and PD mice (Figure 3). Therefore, the reduced impulse derived from MnRN serotonergic neurons may lead to reduced cAMP/CREB signaling in the hippocampal DG without affecting the expression levels of 5-HT_4_R, resulting in the facilitation of the memory extinction.

Adult neurogenesis in the hippocampal DG is required for the establishment and maintenance of remote contextual fear memory [27]. Increased p-CREB levels are known to be associated with neural plasticity and hippocampal neurogenesis [28]. Moreover, CREB signaling plays a critical role in memory extinction and strengthening upon retrieval [29]. Because fear memory extinction occurs in a retrieval-dependent manner, the role of hippocampal DG in memory retrieval could be important for fear memory extinction. The hippocampal DG is composed of many different types of cells, not only matured neural cells, but also immature neuronal and grail cells [30]. Figure 6 shows that the majority of p-CREB positive cells do not express NeuN, suggesting that p-CREB positive cells are either non-neural or immature neural cells; however, this remains unclear. It has recently been reported that increases in adult hippocampal neurogenesis improves memory formation and long-term memory via the regulation of memory extinction [27], and that immature neurons in the hippocampal DG affect the maintenance of hippocampus-dependent memory [31]. However, the relationship between immature neurons and the extinction of fear memory remains unclear. To gain a better understanding of the significance of p-CREB positive cells in memory extinction, further studies aiming to identify the types and roles of cells are necessary.

Overall, the results presented here show that 5-HT4R agonists could be potentially useful as therapeutic drugs for treating cognitive deficits in PD. However, the etiology and onset mechanisms of the cognitive deficits in PD are not well understood. Based on the present study, we proposed the model for cognitive impairment in PD mice and the predicted neural circuit responsible for it (Figure 7). We think that to verify our proposed model experimentally is critical for understanding the physiological role of serotonergic neurons in the MnRN and 5-HT4R in hippocampal learning and memory, which might lead to elucidation of the onset mechanisms of the cognitive deficits in PD. We hope that our study may contribute to the development of a novel therapeutic drug for treating the cognitive deficits in PD.

## 4. Materials and Methods 

### 4.1. Animals and MPTP Treatment

Male C57BL/6 mice (7–8 weeks old) were maintained under controlled temperature (22 ± 2 °C) and humidity (35 ± 5%) on a 12-h light/12-h dark cycle (lights on at 07:00) and allowed ad libitum access to pellet food (Crea Japan, Tokyo, Japan) and water. All procedures were performed in accordance with the Guiding Principles for the Use of Animals in Toxicology in 1989 and approved by the Animal Research Committee at Obihiro University of Agriculture and Veterinary Medicine (approval number: 19-2, Date: 01 April 2019). The animals were humanely euthanized with an overdose of the anesthetic ether at the end of the experiments.

MPTP (Sigma-Aldrich, Tokyo, Japan)-treated PD model mice were prepared as described in our previous report [8]. Eight-week-old mice were given four intraperitoneal injections of a single dose of 20 mg/kg (in 100 µL) in the first two injections and 15 mg/kg (in 100 µL) in the last two injections every 2 h. Saline was administered in a similar manner as a control. Subsequent experiments were conducted 7 days after the last MPTP or saline injection (Figure 8). This MPTP injection protocol can retain a significant reduction in the number of TH-positive cells in the SNpc at least until 16 days after MPTP injection.

### 4.2. Contextual Fear Conditioning Test

All procedures were performed as described in our previous report [5,6]. Mice were trained and tested in conditioning chambers (17.0 × 17.0 × 14.5 cm) that had a stainless steel rod floor through which foot shocks could be delivered (ST-10; Melquest, Toyama, Japan), and all behavior was recorded by an overhead color charge-coupled device (CCD) camera. Mice were submitted to three experimental phases: conditioning, training, and testing. We evaluated freezing behavior during each of these phases, which was defined as a complete absence of movement lasting for longer than 1 s except for respiration and heartbeat. As an index of memory, percentage of time spent freezing was measured during the test session: time spent freezing/total time × 100. During the conditioning phase, as the conditioned stimulus (CS), mice were placed in the chamber, and as the unconditioned stimulus (US), two foot shocks were delivered (2-s duration, 2 mA). Mice were returned to their home cage 30 s after the final foot shock. Next, mice underwent extinction training twice every 24 h after conditioning. In this training, mice were re-exposed to the CS for 30 min without any US, and the percentage of time spent freezing during the initial 3 min of the entire duration of exposure was assessed. Then, 24 h after the last extinction training, the mice were re-exposed to the CS for 3 min, and the percentage of time spent freezing was assessed (extinction test) (Figure 8).

### 4.3. cAMP Assay

cAMP levels were determined using a Cyclic AMP EIA kit (Cayman Chemical Co., Ann Arbor, MI, USA). The hippocampus was collected immediately after the second extinction training (Figure 8). The samples were homogenized in 10 volumes of 5% trichloroacetic acid (TCA) in water on ice using a polytron-type homogenizer (Microtrc Co., Chiba, Japan). Next, the sample was centrifuged at 1500× *g* for 10 min, and the supernatants were collected. TCA was extracted from the sample using water-saturated ether. After removal of the top ether layer, the sample was heated at 70 °C for 5 min to remove the residual ether. The samples were subjected to an enzyme-linked immunosorbent assay according to the manufacturer’s protocol. The absorbance was measured at a wavelength of 415 nm. cAMP levels are shown as pmol/mg of TCA-precipitated tissue. The tissue pellet precipitated by TCA was washed once with an ethanol/ether (1:3) solution, heated at 70 °C for 3 min to remove the residual ethanol/ether, and weighed.

### 4.4. RNA Extraction and Real-Time Quantitative Polymerase Chain Reaction (RT-qPCR) Assay

Total RNA was extracted from hippocampal tissue using Direct-zolTM RNA MiniPrep (Zymo Research, Tustin, CA, USA) according to the manufacturer’s instructions, and quantified by measuring the absorbance at 260 nm. RNA was amplified using the MyGo Mini Real-Time PCR system (IT-IS Life Science, Ltd., Cork, Ireland). One-step RT-qPCR was performed using the MyGo Green 1-step Low Rox (IT-IS Life Science, Ltd.) for a total volume of 20 µL and a template concentration of 10 pg/µL total RNA, according to the manufacturer’s recommendations. The thermal cycling conditions were 45 °C for 10 min as an RT step and 95 °C for 2 min, followed by 40 cycles of 95 °C for 10 s and 60 °C for 20 s. An automatic melting temperature analysis was performed to ensure a single amplified product at the end of the reaction (melting from 60 °C from 95 °C at a ramp rate of 0.1 °C/s). The relative quantification (fold change) of mRNA expression was estimated by the use of the 2^−ΔΔCt^ method [32] as follows: Relative mRNA expression was defined as 2^−Δ(ΔCt)^, where ΔCt = Ct_TARGET_ – Ct_Housekeeping gene_ and Δ(ΔCt) = ΔCt_treated_ groups – ΔCt_CS group_, and Ct_Housekeeping gene_ is the average of the Ct values of β-actin, which was used as the housekeeping gene for each sample to normalize the targeted gene expression.

### 4.5. Fluorescent Double Immunohistochemistry

Thirty minutes after the second extinction training (Figure 8), the mice were transcardially perfused with ice-cold phosphate-buffered saline (PBS) followed by 4% neutral-buffered paraformaldehyde solution under isoflurane anesthesia. The brains were dissected and post-fixed in 4% neutral-buffered paraformaldehyde solution. After fixation, the brains were cut into consecutive 50-µm-thick sections using an oscillating tissue slicer (Linear Slicer Pro7; Dosaka, Kyoto, Japan). The sections were then permeabilized with 0.5% (v/v) Triton X-100 in PBS (T-PBS) for 1 h, blocked in goat serum solution (1:100 in T-PBS) for 1 h, and incubated for 48 h at 4 °C in rabbit anti-phosphorylated CREB (p-CREB) polyclonal antibody (1:800 in T-PBS; Cell Signaling Technology Japan, KK., Tokyo, Japan). The sections were rinsed in T-PBS and incubated for 2 h at 4 °C in Alexa Fluor^®^ 568 goat anti-rabbit IgG (1:5000 in T-PBS; Thermo Fisher Scientific, Waltham, MA, USA). The sections were then incubated with mouse anti-NeuN monoclonal antibody (1:5000 in T-PBS) for 24 h at 4 °C, followed by incubation with Alexa Fluor^®^ 488 goat anti-mouse IgG (1:5000 in T-PBS; Thermo Fisher Scientific) for 2 h at 4 °C. After rinsing in PBS, the sections were mounted and coverslipped with mounting medium suitable for fluorescence microscopy (Vectashield; Vector Laboratories, Inc., Burlingame, CA, USA). The total number of p-CREB-positive cells from both the left and right hippocampal dentate gyrus (DG) in each section was counted using the images obtained by confocal laser-scanning microscopy (C2+; Nikon, Tokyo, Japan). For each mouse, the mean number was calculated by averaging the number of p-CREB-positive cells counted from three sections (positioned 1.82–2.02 mm posterior to the bregma).

### 4.6. Fluorogold (FG) Injection into The MnRN

Mice were anesthetized with anesthesia containing medetomidine (3 µg/10 g body weight, i.p.; Zenoaq, Koriyama, Japan), midazolam (40 µg/10 g body weight, i.p.; Astellas Pharma Inc., Tokyo, Japan), and butorphanol (50 µg/10 g body weight, i.p.; Meiji Seika Pharma, Tokyo, Japan), and placed in a stereotaxic apparatus. FG (Setarch Biotech, LLC, Eugene, OR, USA) was dissolved in sterile saline to make a 2% FG solution, and 0.1 L of the FG solution was injected into the MnRN (4.5 mm posterior to the bregma and 4.7 mm below the surface of the skull) using a 10-µL microsyringe (Ito, Fuji, Japan). Three days after FG injection, the mice were intracardially perfused with ice-cold PBS followed by 4% paraformaldehyde in PBS under isoflurane anesthesia. The brains were post-fixed and sectioned (40-µm thick) as described above. In case of further analysis of fluorescent immunohistochemistry, the sections were then permeabilized with 0.5% (v/v) Triton X-100 in PBS (T-PBS) for 1 h, blocked in Block Ace (1:100 in T-PBS; KAC Co., Kyoto, Japan) for 1 h, and incubated for 24 h at 4 °C in mouse anti-tyrosine hydroxylase (TH) (1:5000 in T-PBS; Merck Millipore, Burlingame, CA, USA), mouse anti-glutamic acid decarboxylase 67 (GAD67) (1:5000 in T-PBS; Sigma-Aldorich, St. Louis, MO, USA), or mouse anti-tryptophan hydroxylase (TPH) (1:5000 in T-PBS; Merck Millipore). The sections were rinsed in T-PBS and incubated for 2 h at 4 °C in Alexa Fluor^®^ 568 goat anti-mouse IgG (1:5000 in T-PBS; Thermo Fisher Scientific). The sections were rinsed in PBS, and then, mounted with mounting medium (Vectashield; Vector Laboratories) and coverslipped. Images were analyzed by a confocal laser-scanning microscope (C2+; Nikon).

### 4.7. Administration of the 5-HT_4_ Serotonin Receptor Agonists

Mice were given an intraperitoneal injection of prucalopride (Sigma-Aldrich) at a dose of 1.5 or 3 mg/kg, or velusetrag (Axon Medchem BV, Groningen, The Netherlands) at a dose of 3 mg/kg in saline containing 1% dimethyl sulfoxide (DMSO) or the vehicle only 2 h prior to both extinction training sessions. We chose the dose of prucalopride based on a previous report by Lucas et al. (2007) [15]. To compare the effects of prucalopride and velusetrag, we used velusetrag at the same dose of prucalopride.

### 4.8. Data Analysis

All statistical analyses were performed using SPSS 16.0 software (SPSS Japan, Inc., Tokyo, Japan). According to the analyses of homoscedasticity using Levene’s test, multiple group comparisons were assessed using one-way analysis of variance (ANOVA), followed by Tukey’s post-hoc test. Statistical differences were considered significant when *p* < 0.05.

## 5. Conclusions

We found that 5-HT_4_R agonists restored the facilitation of contextual fear extinction in PD mice by stimulating the cAMP/CREB pathway in the hippocampal DG. Moreover, GABAergic neurons in the SNr, but not DAergic neurons in the SNpc, were found to project to serotonergic neurons in the MnRN, which are known to project their nerve terminals to the hippocampus. Therefore, the MPTP-induced degeneration of SNpc DAergic neurons might affect GABAergic neurons in the SNr close to the SNpc and influence serotonergic neurons in the MnRN because of the reception of GABAergic neurons from the SNr, thereby resulting in hippocampal dysfunction. These findings suggest that 5HT4R agonists could be potentially useful as therapeutic drugs for treating cognitive deficits in PD by improving the cAMP/PKA/CREB signaling pathway in the hippocampal DG.

## Figures and Tables

**Figure 1 ijms-20-05340-f001:**
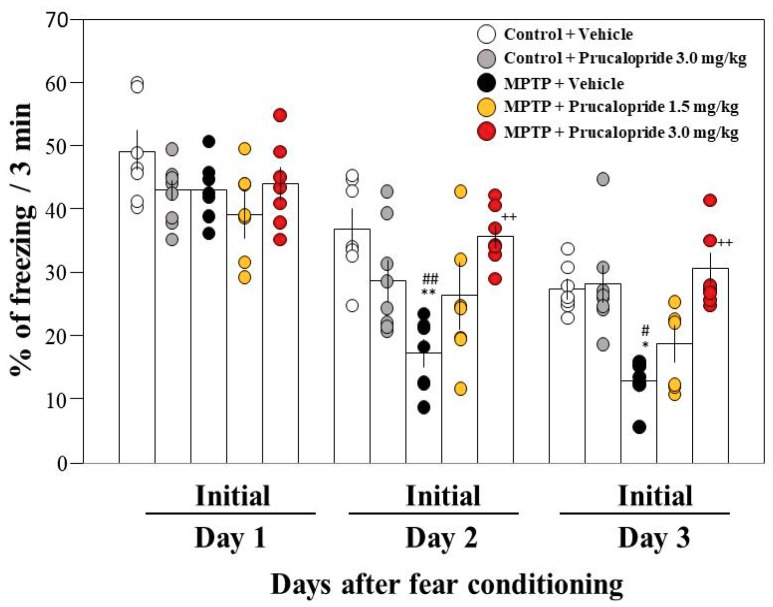
Effects of prucalopride on contextual fear extinction in mice. Mice were given an intraperitoneal injection of prucalopride at a dose of 1.5 or 3.0 mg/kg in saline containing 1% dimethyl sulfoxide (DMSO) or the vehicle only 2 h prior to both extinction training sessions. Data are expressed as the mean ± SEM; *n* = 7–8. * *p* < 0.05 and ** *p* < 0.01 vs. control + vehicle on each day, ^++^
*p* < 0.01 vs. MPTP + vehicle on each day, ^#^
*p* < 0.05 and ^##^
*p* < 0.01 vs. control + prucalopride on each day (one-way analysis of variance (ANOVA) followed by Tukey’s post-hoc test). MPTP: 1-methyl-4-phenyl-1,2,3,6-tetrahydropyridine.

**Figure 2 ijms-20-05340-f002:**
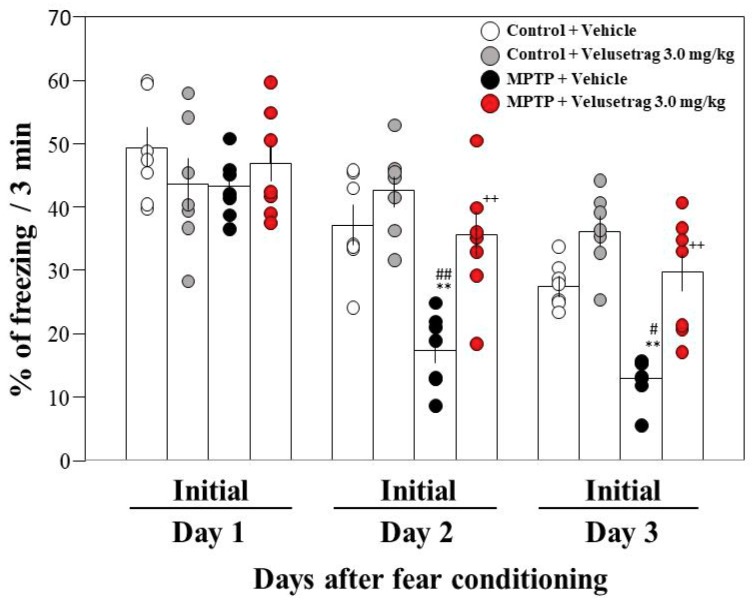
Effects of velusetrag on contextual fear extinction in mice. Mice were given an intraperitoneal injection of velusetrag at a single dose of 3.0 mg/kg in saline containing 1% DMSO or the vehicle only 2 h prior to both extinction training sessions. The data of the control + vehicle and MPTP + vehicle are the same data shown in Figure 1. Data are expressed as the mean ± SEM; *n* = 7–8. ** *p* < 0.01 vs. control + vehicle on each day, ^++^
*p* < 0.01 vs. MPTP + vehicle on each day, ^#^
*p* < 0.05 and ^##^
*p* < 0.01 vs. control + velusetrag on each day (one-way ANOVA followed by Tukey’s post-hoc test).

**Figure 3 ijms-20-05340-f003:**
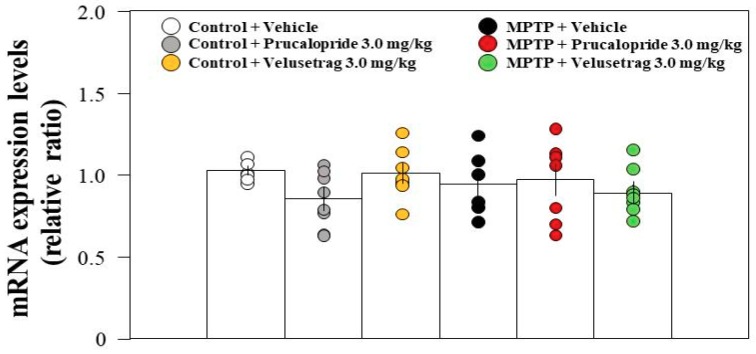
RT-qPCR analysis of the hippocampal mRNA expression levels of 5-HT_4_ receptor (5-HT_4_R) in vehicle-administered control and PD mice, and in prucalopride or velusetrag-administered control and Parkinson´s disease (PD) mice. Data are expressed as the mean ± SEM: *n* = 6–8 per group. No significant differences were observed between groups (one-way ANOVA followed by Tukey’s post-hoc test).

**Figure 4 ijms-20-05340-f004:**
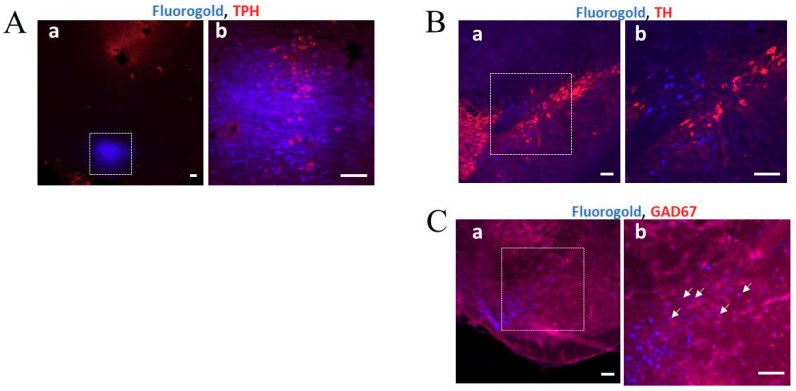
Retrograde labeling of neurons following fluorogold (FG) injection into the median raphe nucleus (MnRN). (**A**) The square on the confocal laser-scanning microscope images under low (a) and high (b) magnification indicates the MnRN region at 3 days after FG injection. The image shows that FG (blue) was precisely injected into the MnRN. Tryptophan hydroxylase (TPH)-positive cells (red) were observed in the MnRN. Scale bar: 100 µm. (**B**) and (**C**) The square on the photomicrograph taken under low (a) and high (b) magnification indicates the substantia nigra pars compacta (SNpc) (**B**) and the reticular part of the reticular part of the substantia nigra (SNr) (**C**), which were analyzed using a confocal laser-scanning microscope at 3 days after FG injection. Scale bar: 100 µm. FG-labeled cells (blue) (**B**-a and -b, C-a and -b) and GAD67-positive cells (red) (**C**-a and -b) were co-localized in the SNr regions (indicated by white arrows) (**C**-b), but TH-positive cells (red) (**B**-a and -b) were not co-localized in the SNpc regions.

**Figure 5 ijms-20-05340-f005:**
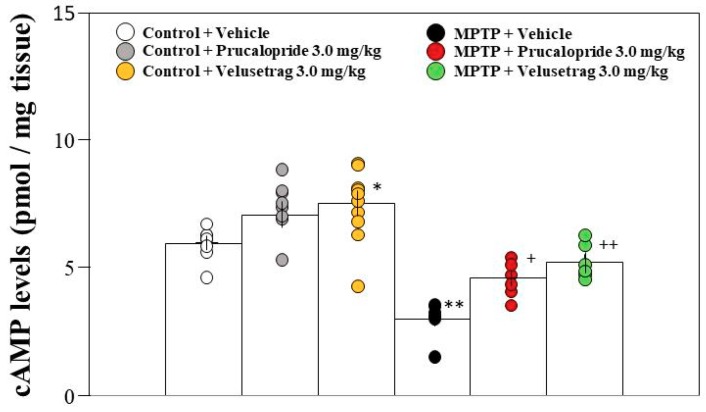
Administration of the 5-HT_4_R agonists to PD mice restored the decrease in cyclic adenosine monophosphate (cAMP) levels in the hippocampus. Mice were given an intraperitoneal injection of prucalopride or velusetrag at a single dose of 3.0 mg/kg in saline containing 1% DMSO or the vehicle only 2 h prior to both extinction training sessions. The hippocampus was collected immediately after the second extinction training. cAMP levels are shown as pmol/mg of trichloroacetic acid (TCA)-precipitated tissue. Data are expressed as the mean ± SEM; *n* = 7–8. * *p* < 0.05 and ** *p* < 0.01 vs. control + vehicle, ^+^
*p* < 0.05 and ^++^
*p* < 0.01 vs. MPTP + vehicle (one-way ANOVA followed by Tukey’s post-hoc test).

**Figure 6 ijms-20-05340-f006:**
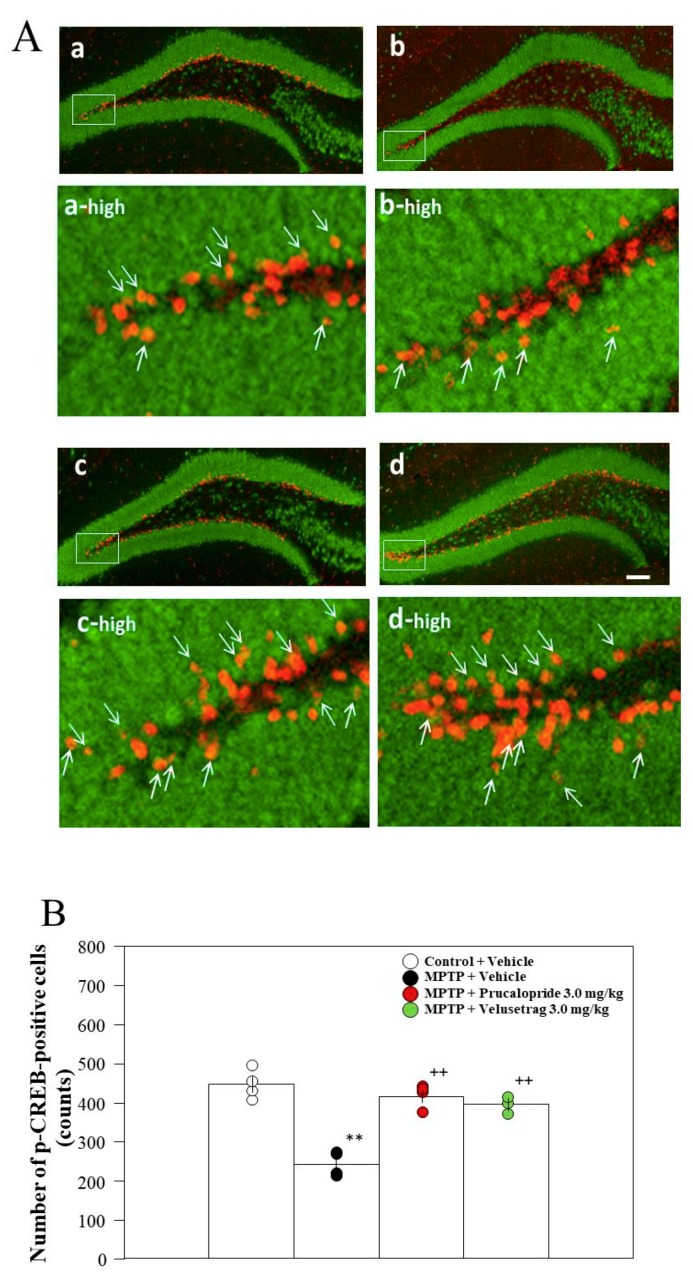
Analysis of the number of p-CREB-positive cells in the dentate gyrus (DG) of the hippocampus after extinction training. (**A**) Immunohistochemistry for p-CREB (red) and NeuN (green) in the DG after fear extinction. (a) Control + vehicle, (b) MPTP + vehicle, (c) MPTP + prucalopride 3.0 mg/kg, (d) MPTP + velusetrag 3.0 mg/kg. Scale bar = 100 µm. The images of “a-high”, “b-high”, “c-high”, and “d-high” were partially expanded from the images of the white dotted box area in a, b, c, and d, respectively. Some p-CREB-positive cells co-localize with NeuN (yellowish red, indicated by white arrows). (**B**) The number of p-CREB-positive cells in the DG. The mean was calculated by averaging the number of p-CREB-positive cells counted from three 50-µm sections (obtained from a position 1.82–2.02 mm posterior to the bregma). Data are expressed as the mean ± SEM; *n* = 3–4 per group. ** *p* < 0.01 vs. control + vehicle, ^++^
*p* < 0.01 vs. MPTP + vehicle (one-way ANOVA followed by Tukey’s post-hoc test).

**Figure 7 ijms-20-05340-f007:**
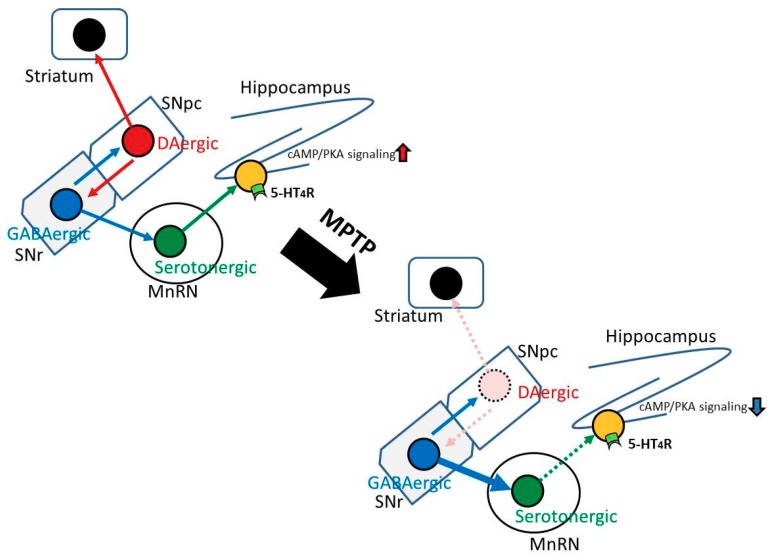
The proposed model for cognitive impairment in PD mice and the predicted neural circuit responsible for it. The degeneration of SNpc dopaminergic (DAergic) neurons might affect γ-aminobutyric acid-ergic (GABAergic) neurons in the SNr close to the SNpc, which in turn, may influence serotonergic neurons in the MnRN receiving GABAergic neurons from the SNr, resulting in hippocampal dysfunction.

**Figure 8 ijms-20-05340-f008:**
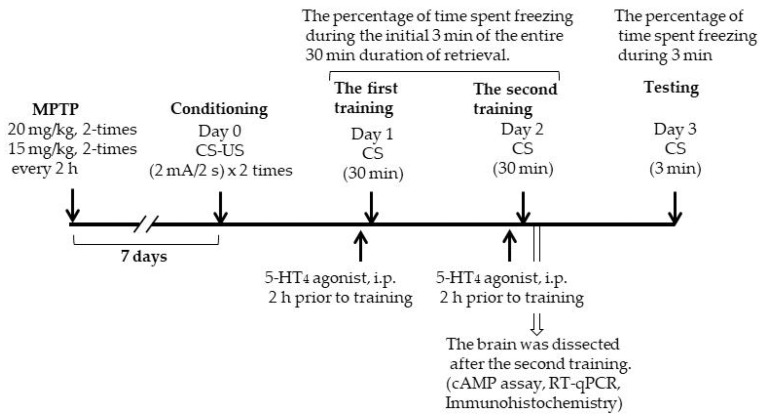
A schematic figure explaining the planning of the experiments.

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
