# Peer review of "Serotonin 5-HT4 Receptor Agonists Improve Facilitation of Contextual Fear Extinction in an MPTP-Induced Mouse Model of Parkinson’s Disease"

_ijms, 2019, doi:10.3390/ijms20215340_

Round 1

Reviewer 1 Report

The manuscript by Ishii et al., examines whether 5-HT4 eeceptor agonists improve the facilitation of contextual fear extinction in mice that have been treated with MPTP to induce a form of Parkinson’s.  The results do show a reversal of the MPTP effect by the agonists and the authors further demonstrate that cAMP levels are elevated by the agonists.  While the results are promising, there are some items that would strengthen the study.      

1. Are the control + vehicle and MPTP + vehicle the same exact mice and data in Figures 1 and 2? If so, this should be made clear in the text. 

2. Could the effect of the 5HT R agonists be at the level of translation rather than transcription? If so, a western blot should be performed.

3. In Figure 4, it should be indicated what the while arrows are pointing to.

4. In Figure 5, are the MPTP + prucalopride and MPTP + velusetrag statistically different from control + vehicle? If so, they should have an asterisk to indicate they have not been rescued all the way back to control levels. 

5. In Figure 6, the low power images provided do not allow the reader to see if these is co-localization of NeuN and p-CREB as suggested in the Discussion (line 194).

6. The statement in the Discussion on lines 170-171 should be softened since no causation between the increased p-CREB positive cells and improved cognitive function in mice is demonstrated in this particular study. Although the authors’ previous paper demonstrates that increased cAMP signaling restores facilitation of memory extinction in PD mice, the true demonstration that CREB signaling is directly involved in this process would be to block CREB signaling and then prevent the effect on behavior as induced by the 5HT agonists and/or rolipram.

7. Rather than using solid bar graphs, the authors should show the data points as scatter dots inside a clear bar graph with the standard error. This would demonstrate the distribution of the data better.

8. It would be helpful to include a figure that shows the proposed model for the various inputs from the SNpc and SNr to mnRN and hippocampus.

9. The Discussion is quite brief and could go into more analysis of the data itself.

Author Response

Responses to reviewer #1 comments

1. We appreciate your comments and question. The data of the control + vehicle and MPTP + vehicle shown in Fig.1 and Fig.2 were obtained from the same mice. In response, we have revised and added the sentence in the Fig.2-legend to mention about this issue (page 4, line 91). Page 4, line 91 now read “The data of the control + vehicle and MPTP + vehicle are the same data shown in Fig.1.”

2. Thank you for your question. Although the effect of 5-HT4R agonists should be caused by unknown signal proteins at the translational level, we have not determined them yet. We, therefore, could not perform a western blot analysis in the present study. We are now trying to find key signal protein molecules induced by the CREB-dependent manner.

3. Thank you for your comment. In response, we have revised and added the sentence in the Fig.4-legend to mention about this issue (page 5, line 118). Page 5, line 118 now read “FG-labeled cells (blue) (B-a and -b, C-a and -b) and GAD67-positive cells (red) (C-a and -b) were co-localized in the SNr regions (indicated by white arrows) (C-b), but TH-positive cells (red) (B-a and -b) were not co-localized in the SNpc regions.”

4. Thank you for your question. Neither the MPTP + prucalopride nor the MPTP + velusetrag was statistically different from the control + vehicle.

5. We greatly appreciate your comments. In response, we have newly added a partially expanded image (a-high, b-high, c-high, and d-high) to allow the reader to see co-localization of NeuN and p-CREB more clearly in Fig.6. We have also revised and added the sentence about this issue in the section of Fig.6 legend (page 8, line 150). Page 8, line 150 (in Fig.6 legend) now read “The images of a-high, b-high, c-high, and d-high were partially expanded from the images of a, b, c, and d, respectively. Some p-CREB-positive cells co-localize with NeuN (yellowish red, indicated by white arrows).”

6. We greatly appreciate your comments and agree with your suggestion. The corresponding statement in the Discussion was softened in the revised manuscript. In response, we have revised the sentences about this issue in the section of Discussion (page 8, line 177). Page 8, line 177 now read “These findings suggest that reduced cAMP/CREB signaling in the DG might be involved in a cause of cognitive impairment in PD mice.”

7. Thank you for your comment. In response, we have revised all figures using solid bar graphs (Fig. 1, 2, 3, 5, and 6B) and showed the data points as scatter dots inside a clear bar graph with the standard error in the revised graphs.

8. In response, we have newly added a figure (Figure 7) that shows the proposed model for cognitive impairment in PD mice and the predicted neural circuit responsible for it in the section of Discussion.

9. In response, we have added the sentences about the effects of velusetrag on contextual fear extinction and hippocampal cAMP levels in the control mice in the section of Discussion (page 8, line 179). Page 8, line 179 now read “On the other hand, administration of velusetrag (3.0 mg/kg) to control mice significantly increased hippocampal cAMP levels (Figures 5), but did not affect contextual fear extinction, and maintained it (Figures 2). These results suggest that cAMP contents higher than a certain threshold level in the hippocampal neurons might be necessary for normal extinction.”                                                                                                                             

Reviewer 2 Report

The authors aim to use MPTP induced PD mouse model to test if manipulating 5HT Receptors can affect PD mouse model behaviorally. The discovery is 5HT agonists can in fact restore contextual learning. 

Could I ask the authors to clarity the last figure, which compared with with is significant on the figure, in addition to the descriptions below? PD mouse model also have motor defects, after applying the agonists, are motor skills affected or improved?

Thanks. 

Author Response

Thank you for your question and comment. In response, we have newly added a partially expanded image (a-high, b-high, c-high, and d-high) to allow the reader to see more clearly in Fig.6. We have also revised and added the sentence about this issue in the section of Fig.6 legend (page 7, line 148). Page 7, line 148 (in Fig.6 legend) now read “(A) Immunohistochemistry for p-CREB (red) and NeuN (green) in the DG after fear extinction. (a) Control + Vehicle, (b) MPTP + Vehicle, (c) MPTP + Prucalopride 3.0 mg/kg, (d) MPTP + Velusetrag 3.0 mg/kg. Scale bar = 100 mm. The images of a-high, b-high, c-high, and d-high were partially expanded from the images of a, b, c, and d, respectively. Some p-CREB-positive cells co-localize with NeuN (yellowish red, indicated by white arrows).”

Next, to examine motor deficits in PD mice, we have used the rotarod test that is a useful method for detecting hypokinesia. When we measured the duration that mice stayed on the rotating rod at a certain speed and compared its rotarod performance in control and PD mice, the latency to fall from the rod in PD mice was significantly decreased compared with control mice (please see our previous report: Kinoshita et al. (2015) Life Sci. 137, 28-36.). Administration of the 5-HT4R agonist prucalopride and velusetrag to PD mice, however, did not improve the impaired rotarod performance (data not shown). In response, we have added the sentences about this issue in the section of Result (page 2, line 81). Page 2, line 81 now read “On the other hand, the two 5-HT4R agonists did not improve the impaired rotarod performance in PD mice (data not shown).”

Reviewer 3 Report

The study of models of Parkinson’s disease (PD) and the molecular targets that modulate this pathology is without any doubt essential in the search of new treatments aim to prevent damage and fight against the development of the pathophysiology of PD. For this reason, this article presents relevant information. The topic is correctly addressed and enough clear. The methodology is well address and the bibliography is correctly updated. It is worth highlighting the quality of the figures.  However, some points should be clarified. 

1. Formal points:

Some abbreviations should be added to the list of abbreviations, for example DG. Statistic data analysis should be specify in each figure, for example are how researchers made the multiple group comparison test?, with or without repeated measures?, which statistic test was used?. Regarding method, it could be useful a schematic figure with scheme explaining the planning of the experiments. Otherwise, it is difficult to understand the timing of the experiments, and how animals were distributed, for each experiment. At acknowledgments paragraph authors should rewrite this section with the data of this study.

Content points:

2. Authors described at Materials and Methods that administered 5-HT4 serotonin receptors agonists (Prucalopride and Velusetrag). They described the doses, but did not explain why they chose these doses and why they gave the administration of the same agonists in 4 phases (paragraph 4.1and 4.7)

3. Why authors do not study the levels of  DA, 5-HTP with HPLC?, because it can be useful for explaining their results.

4. Authors mentioned that both 5-HTP4R agonists restored facilitation of contextual fear extinction in PD mice by stimulating the cAMP/CREB pathway in the dentate gyrus of the hippocampus. Authors also studied cAMP levels in the hippocampus and number of p-CREB-positive cells in the hippocampus of the DG. Therefore, it could be also useful to study by western blot the mentioned signaling pathway, analyzing levels of CREB, p-CREB, and therefore activation of the pathway. If authors considered that it is not necessary they should explain it.

5. The justification is well address, but in order to improve the quality of the article it could be useful to explain the perspective and future directions, which are the specific future questions to be answer? What kind of information is needed, in which models, in order to could use this information, for finding specific translational solutions? Authors should explain this point better, in a brief way.

Author Response

1. We greatly appreciate your comments. In response, some other abbreviations have been added to the list of abbreviations. We have specified statistic analysis in each figure and newly added a schematic figure (Figure 8) with explaining the planning of the experiments in the section of Materials and Methods (page 11, line 306 - 307). We have also corrected the acknowledgments paragraph.

2. We greatly appreciate your comments and questions. We chose the dose of prucalopride based on a previous report by Lucas et al. (Neuron 2007, 55, 712-725.). On the other hand, velusetrag was used at the same dose of prucalopride to compare the effects of prucalopride and velusetrag. In response, we have added the report by Lucas et al. as the number of [33] in the list of references, and also revised and added the sentences about this issue in the section of Materials and Methods (page 12, line 331). Page 12, line 331 now read “We chose the dose of prucalopride based on a previous report by Lucas et al. (2007) [33]. To compare the effects of prucalopride and velusetrag, we used velusetrag at the same dose of prucalopride.”

For our MPTP injection protocol (Eight-week-old mice were given four intraperitoneal injections of a single dose of 20 mg/kg (in 100 mL) in the first two injections and 15 mg/kg (in 100 mL) in the last two injections every 2 h), because this protocol can retain a significant reduction in the number of tyrosine hydroxylase (TH)-positive cells in the SNpc at least until 16 days after MPTP injection, we utilized the MPTP protocol for producing PD mice. In response, we have revised and added the sentences about this issue in the section of Materials and Methods (page 10, line 240). Page 10, line 240 now read “This MPTP injection protocol can retain a significant reduction in the number of TH-positive cells in the SNpc at least until 16 days after MPTP injection.”

3. Thank you for your question. We agree with your idea and realize the importance to measure the levels of DA or 5-HT. MPTP causes selective loss of DAergic neurons only in the SNpc, and, in the present study, the damage to DAergic neurons in the SNpc was not recovered even after administration of 5-HT4R agonist. We, therefore, did not examine the levels of DA in this study. On the other hand, MPTP does not directly damage serotonergic neurons but the degeneration of the SNpc DAergic neurons might lead to the hypofunction of serotonergic neurons in the MnRN, resulting in hippocampal dysfunction. Therefore, we think that an assay of the 5-HT levels using HPLC would be necessary for further study to investigate the physiological role of serotonergic neurons in the MnRN and 5-HT4R in hippocampal learning and memory.

4. We appreciate your comments. The number of p-CREB-positive cells in the hippocampus of PD mice decreased only in the DG but not in hippocampal other areas such as CA1 and CA3. Moreover, the majority of p-CREB positive cells in the DG did not express NeuN, suggesting that p-CREB positive cells are either non-neural or immature neural cells. In the present study, we considered that it should be important to show those distinctive characteristic features of p-CREB-expressed cells. Therefore, we performed the analysis of fluorescent double immunohistochemistry, which can determine not only the number of p-CREB-positive cells but also the cell type (i.e., neuron and non-neuron or immature neural cells) of p-CREB-expressed cells, instead of western blot analysis. That is the reason why we did not choose the western blotting analysis of CREB and p-CREB in the present study.

5. We appreciate your comments. In response, we have newly added a figure (Figure 7) that shows the proposed model for cognitive impairment in PD mice and the predicted neural circuit responsible for it in the section of Discussion. We have also added the sentences about this issue in the section of Discussion (page 9, line 214). Page 9, line 214 now read “Overall, the results presented here show that 5-HT4R agonists could be potentially useful as therapeutic drugs for treating cognitive deficits in PD. However, the etiology and onset mechanisms of the cognitive deficits in PD are not well understood. Based on the present study, we proposed the model for cognitive impairment in PD mice and the predicted neural circuit responsible for it (Figure 7). We think that to verify our proposed model experimentally is critical for understanding the physiological role of serotonergic neurons in the MnRN and 5-HT4R in hippocampal learning and memory, and also might lead to elucidation of the onset mechanisms of the cognitive deficits in PD. We hope that our study may contribute to the development of a novel therapeutic drug for treating the cognitive deficits in PD.”

Round 2

Reviewer 3 Report

Authors have answered all questions properly. All changes were added to the manuscript. Therefore, I have no other comments.